# Secure Dynamic Event-based Consensus for Networked Multi-agent Systems subject to Distributed DoS Attacks

Bohan Li, Qing Gao✉, Zhenqian Wang, Wei Wang, and Jinhu Lü

*School of Automation Science and Electrical Engineering, Beihang University, Beijing 100191, China*

gaoqing@buaa.edu.cn;

*Abstract*—This paper addresses the secure leader-following consensus problem in networked multi-agent systems (MASs) subjected to distributed DoS attacks. We propose a robust consensus protocol, leveraging dual-terminal dynamic event-triggering mechanisms (DETMs), to ensure secure consensus in compromised systems. Attackers can launch asynchronous, aperiodic DoS attacks on different communication channels within the MAS. The validity of a distributed DoS attack is defined by its ability to disrupt the communication topology's connectivity. By implementing two DETMs in each agent, both communication and control update frequencies are reduced, thereby conserving control energy and communication resources. Additionally, we introduce a secure hybrid update law to maintain secure consensus under distributed DoS attacks. Theoretical analysis confirms that the proposed protocol guarantees asymptotically stable leader-following consensus, given that the duration and frequency of valid attacks are constrained, and no agent exhibits Zeno behavior. Numerical simulations are provided to demonstrate the effectiveness of the proposed protocol.

*Index Terms*—Secure cooperation, distributed DoS attack, networked multi-agent system (MAS), dynamic event-based control.

## I. INTRODUCTION

In recent years, with the advancement of information and communication technologies, control systems have progressively evolved towards larger scales and more complex structures. Consequently, distributed control, characterized by its ability to enhance flexibility, scalability, robustness, and fault tolerance, has found widespread application across various domains such as sensor networks [1], microgrids [2], and production systems [3]. In reality, as a critical subset of cyber-physical systems characterized by inter-agent communication and collaboration, MASs are susceptible to adversarial cyber-physical attacks such as Denial of Service (DoS) and deception attacks [4].A common type of DoS attack within networked MASs involves overwhelming the communication channels with excessive traffic or selectively blocking messages, thereby jamming the communication among agents. These attacks can lead to failures in achieving consensus, coordination, and other collective behaviors essential for the system's operation, as timely and reliable communication is crucial for distributed control of networked MASs. Hence, a focal concern lies in designing a resilient distributed control protocol to ensure the secure consensus of multi-agent systems that are susceptible to such attacks.

Up to now, numerous studies have focused on investigating attack modeling and mitigation methods for MASs subject to DoS attacks. The existing literature on secure cooperative control of multi-agent systems under DoS attacks can be divided into two major categories. The first category assumes that the DoS attack follows some kind of probability distribution, such as the Bernoulli-distributed attack model ( [5], [6]) and the Markov-switching attack model ( [7], [8]). The authors in [5] propose a distributed output-feedback control strategy for heterogeneous linear MASs subject to random DoS attacks and aperiodic sampling to achieve robust leader-following output consensus. In this research, the attacked closed-loop system is modeled as a discrete-time switched stochastic delay system. In [7], the secure leader-following consensus for linear MASs under strategic DoS attacks is studied, where the attack is described as a random Markov process. A distributed secure consensus protocol is designed to achieve mean-square exponential consensus tracking. The observer-based distributed secure consensus problem for attacked MASs is investigated in [8], where the MAS under DoS attacks is modeled as a MAS with Markov switching communication topology. An observer-based secure consensus strategy using output information is proposed to guarantee the consensus of the agents' states. The second category assumes that the frequency and duration of the DoS attacks are constrained, as the energy for launching attacks is limited in reality [9]–[11]. The researchers in [9] analyze a single integrator MAS under asynchronous attack and prove that if the frequency and duration of the attacks on each communication channel are constrained, a DoS-resilient consensus protocol can be designed to ensure secure consensus. Reference [10] proposes a secure static event-triggered consensus protocol for a general linear MAS whose communication channels are attacked synchronously. If the conditions on DoS attack duration and frequency are satisfied, resilient leaderless and leader-following consensus can be achieved with exponential stability. The dynamic event-triggered and self-triggered leader-following consensus for linear MASs under asynchronous DoS attacks are studied in [11]. The concept of a valid attack is introduced, where an attack is considered valid if the connectivity of the communication topology is destroyed. If the frequency and duration of the valid attacks are limited, secure consensus can be ensured.

Based on the above review, although several DoS-resilient

consensus protocols have been proposed, there are still some shortcomings: *i)* References [5]–[8] model the DoS attack as specific probability distributions and random processes. However, the pattern of attacks can be arbitrary and thus cannot be described by any random model. *ii)* Though the event/self-triggered consensus protocol proposed in [10], [11] can avoid continuous communication and reduce communication burden, as a subset of cyber-physical systems, the agents in networked MASs usually employ discontinuous sampling and control updating to save control energy and computing resources. To address these shortcomings, we propose a novel distributed dual-terminal dynamic event-triggered secure consensus protocol to guarantee DoS-resilient leader-following consensus of a linear networked MAS while saving communication resources and control energy. Our key contributions are listed as follows:

1) Compared to [10] and [11], which apply event-based communication mechanisms, our proposed protocol applies dynamic event-triggered mechanisms (DETMs) to both communication and control updates. Thus, continuous control updates can also be avoided. Compared to the static event-triggered mechanism used in [10], the DETMs can further reduce the trigger frequency. Therefore, the communication burden and control update frequency can be further reduced.

2) Compared to [5]–[8], this paper doesn't require the DoS attacks to follow specific random processes and probability distributions. Instead, it only requires certain conditions on valid attack duration and frequency (see Assumptions 2 and 3). In addition, motivated by [11], we model the attacks and design the secure consensus strategy using a connectivity-based approach. Compared to [9] which analyzes the attack duration and frequency on each channel, this approach is more convenient for conducting stability analysis, as it only focuses on the valid attack interval during which the connectivity of the communication topology is destroyed.

*Organizations:* In Section II, the basic algebraic graph theory, the communication topology and agent dynamics of the investigated networked MASs, the distributed DoS attack model, and control objective are introduced. Section III presents the design and stability analysis of the secure dual terminal dynamic event-triggering leader-following consensus protocol. The secure leader-following consensus simulation is given in Section IV to verify the effectiveness of the designed protocol. Finally, the conclusion of this article is presented in Section V.

*Notations:* In this paper, $\mathbb{R}$ denotes the set of real numbers, specially, $\mathbb{R}_{\geq 0}$ and $\mathbb{R}^{n \times m}$ represent the sets of non-negative real numbers and $n \times m$ real matrices, respectively. $\mathbb{N}$ is the set of natural numbers. Take any $x \in \mathbb{R}$, the notation $x_n$ denotes a column vector with all the $n$ elements being $x$. In order to construct a diagonal matrix with entries $a_i$ and a column vector with entries $c_i$, the notations $\text{diag}\{c_1, \ldots, c_n\}$ and $\text{col}\{a_1, \ldots, a_n\}$ are introduced. The Kronecker product and the Euclidean norm operation are denoted as $\otimes$ and $\|\cdot\|$,

respectively. Given two symmetric matrices $\mathcal{M} \in \mathbb{R}^{n \times n}$ and $\mathcal{N} \in \mathbb{R}^{n \times n}$, the matrix inequality $\mathcal{M} > \mathcal{N}$ implies that $\mathcal{M} - \mathcal{N}$ is a positive definite matrix, and the operations for taking the smallest and largest eigenvalue of $\mathcal{N}$ are defined as $\lambda_{\min}(\mathcal{N})$ and $\lambda_{\max}(\mathcal{N})$, respectively. $\lambda_{\min}(Q)$ ($\lambda_{\max}(Q)$) denotes the smallest (largest) eigenvalue of $Q$. In addition, the operations $\alpha_{\min}(\cdot)$ and $\alpha_{\max}(\cdot)$ indicates calculating the minimum and maximum singular value of a matrix, respectively.

## II. Preliminaries and Problem Formulation

This section first introduces the fundamental algebraic graph theory commonly used in the field of MASs, and then presents the problem formulation and control objectives of this paper.

### A. Algebraic Graph Theory

Let $\mathcal{G} = (\mathcal{V}, \mathcal{E}, \mathcal{A})$ be a directed graph, where $\mathcal{V}$, $\mathcal{E}$ and $\mathcal{A} = (a_{ij}) \in \mathbb{R}^{N \times N}$ are the set of nodes, set of directed edges, and adjacency matrix of $\mathcal{G}$, respectively. In the field of collaborative control, graph $\mathcal{G}$ be applied to describe the communication topology of MASs. Generally speaking, each agent can be viewed as a node $i \in \mathcal{V}$, if agent $i$ can transmit information to agent $j$, then the directed edge $(i, j) \in \mathcal{E}$. The weights of the adjacency matrix $\mathcal{A}$ are defined as $a_{ii} = 0$, $a_{ij} > 0$ if and only if $(j, i) \in \mathcal{E}$, otherwise $a_{ij} = 0$. The income neighborhood set of agent $i$ is represented by $\mathcal{N}_i(\mathcal{G}) = \{j \in \mathcal{V} \mid (j, i) \in \mathcal{E}\}$ while $\bar{\mathcal{N}}_i(\mathcal{G}) = \{j \in \mathcal{V} \mid (i, j) \in \mathcal{E}\}$ denotes the set of its outcome neighbors. In addition, the Laplacian matrix of graph $\mathcal{G}$ is defined as $\mathcal{L} = [l_{ij}] \in \mathbb{R}^{N \times N}$, where $l_{ii} = \sum_{j=1}^{N} a_{ij}$ and $l_{ij} = -a_{ij}$ if $i \neq j$. . A directed path from node $i_1$ to node $i_h$ consist of a sequence of ordered directed edges $(i_k, i_{k+1})$, $k = 1, 2, \ldots, h - 1$. Moreover, in a directed graph $\mathcal{G}$, if there is a node $i$ which can reach all other nodes through a directed path, then this graph is considered to contain a directed spanning tree with $i$ as the root node.

### B. Multi-agent System Model

Consider a linear multi-agent system with $N$ identical agents, the dynamics of agent $i$ is described by

$$\dot{x}_i(t) = Ax_i(t) + Bu_i(t), \quad t \in \mathbb{R}_{\geq 0}, \qquad (1)$$

where $x_i(t) \in \mathbb{R}^n$ and $u_i(t) \in \mathbb{R}^m$, $i = 1, 2, \ldots, N$ denote the state and control input, respectively. $A \in \mathbb{R}^{n \times n}$ and $B \in \mathbb{R}^{n \times m}$ are the state and input matrices of (1), respectively. The leader's state $x_0(t) \in \mathbb{R}^n$ is generated by the following dynamics:

$$\dot{x}_0(t) = Ax_0(t), \qquad (2)$$

where the $A \in \mathbb{R}^{n \times n}$ is identical to the state matrix in (1). The matrix $A$ doesn't have to be Hurwitz in this article, it is required that $(A, B)$ is stabilizable. Under this condition, given any two matrices $Q, R > 0$, the algebraic Riccati equation (ARE) $PA + A^T P - PBR^{-1}B^T P = -Q$ has a positive-definite solution $P > 0$.

In the leader-following consensus problem addressed in this paper, the communication topology among the followers in (1) is represented by directed graph $\mathcal{G} = (\mathcal{V}, \mathcal{E}, \mathcal{A})$, where $\mathcal{V} = \{1, 2, \ldots, N\}$. Meanwhile, graph $\tilde{\mathcal{G}} = (\tilde{\mathcal{V}}, \tilde{\mathcal{E}}, \tilde{\mathcal{A}})$

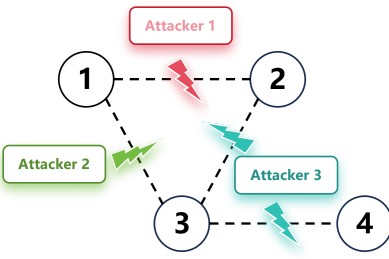

Fig. 1. Example of distributed DoS attack.

characterize the communication topology among both follower and leader agents with $\tilde{\mathcal{V}} = \{0, 1, \ldots, N\}$. Moreover, a diagonal pinning matrix $\mathcal{D}$ can be introduced since the leader agent can be connected to several followers, where $\mathcal{D} = \text{diag}\{d_1, d_2, \ldots, d_N\} \in \mathbb{R}^{N \times N}$ with $d_i > 0$ if $(i, 0) \in \tilde{\mathcal{E}}$, and $d_i = 0$ otherwise. Denote $\mathcal{H} = \mathcal{L} + \mathcal{D}$, where $\mathcal{L}$ represents the Laplacian matrix of $\mathcal{G}$. In this article, the communication graph of the studied MAS should meet

*Assumption 1:* $\tilde{\mathcal{G}}$ contains a directed spanning tree with the leader node as the root node.

*Lemma 1:* [12], [13] Under Assumption 1, $\mathcal{H}$ is a non-singular M-matrix, that is, all the eigenvalues of $\mathcal{H}$ have positive real parts. let $\theta = [\theta_1^{-1}, \ldots, \theta_N^{-1}]^T = \mathcal{H}^{-T} 1_N$, $\Theta = \text{diag}\{\theta_1^{-1}, \ldots, \theta_N^{-1}\}$ is positive-definite, and $\Omega = \Theta\mathcal{H} + \mathcal{H}^T\Theta > 0$.

In previous studies like [14] and [15], distributed control protocol in the following form can be designed such that the above system can achieve leader-following consensus:

$$
\begin{aligned}
u_i(t) &= K\vartheta_i(t), \\
\vartheta_i(t) &= a_{ij}\sum_{i=1}^{N}(x_i(t) - x_j(t)) + d_i(x_i(t) - x_0(t)).
\end{aligned}
\tag{3}
$$

### C. Modelling of DoS attack

DoS attack is a form of cyber assault that disrupts network resources, rendering them inaccessible to legitimate users by inundating the network with excessive, invalid requests or by interfering with communication links. Within the realm of networked MASs, DoS attacks can lead to delays, data loss, or total failure of communication channels between agents. Given that agents in a MAS depend on prompt and dependable communication to exchange information and make collective decisions, such attacks can severely undermine the system's coordination, consensus, and overall functionality.

In this study, the networked MASs are examined under the influence of distributed DoS attacks, wherein the communication channels between various agent pairs are asynchronously disrupted. Fig 1 illustrates a schematic example of distributed DoS attacks.

In this study, we apply a channel-based approach to model distributed DoS attacks. Prior researches like [10], [11], [13]–[15] indicate that designing a secure consensus protocol typically requires maintaining the connectivity of the communication topology (as in Assumptions 1). In the event of a distributed DoS attack, if certain edges are compromised but

the overall graph connectivity remains intact, the protocol can still achieve secure consensus, rendering the attack ineffective. Conversely, if the attack disrupts the connectivity, it is deemed effective.

Based on the above considerations, we define the following two intervals:

$$
\begin{cases}
\mathcal{D}(t_1, t_2) = [t_1, t_2) \bigcap \bigcup_{n \in \mathbb{N}}[h_n, h_n + \tau_n), \\
\mathcal{S}(t_1, t_2) = [t_1, t_2) \backslash U^{(}t_1, t_2),
\end{cases}
\tag{4}
$$

where $h_n$ and $\tau_n$ are the $off/on$ transition and the duration of the $n$th valid DoS attack interval, respectively, $\mathcal{D}(t_1, t_2)$ and $\mathcal{S}(t_1, t_2)$ represent the unhealthy interval and healthy interval over $t \in [t_1, t_2)$.

*Assumption 2 (Attack Duration):* Given $\forall\, t_2 > t_1 \geq 0$, the total duration valid DoS duration satisfies:

$$
|\mathcal{D}(t_1, t_2)| \leq \delta_D + \frac{t_2 - t_1}{T},
\tag{5}
$$

where $\delta_D \in \mathbb{R}_{\geq 0}$ and $T \in \mathbb{R}_{\geq 1}$.

*Assumption 3 (Attack Frequency):* Given $\forall\, t_2 > t_1 \geq 0$, the valid attack frequency is represented by:

$$
\Gamma(t_1, t_2) = \frac{\Lambda(t_1, t_2) - \delta_F}{t_2 - t_1},
\tag{6}
$$

where $\Lambda(t_1, t_2)$ is the total number of valid attacks over $t \in [t_1, t_2)$, and $\delta_F \in \mathbb{R}_{\geq 0}$.

### D. Control Objective

In this study, our goal is to design a dual-terminal dynamic event-triggered mechanisms (DETMs) based protocol, so that the attacked networked MAS introduced in Section II-B and II-C can achieve the following secure leader-following consensus while reducing communication burden and saving control energy:

$$
\lim_{t \to \infty} \|x_i(t) - x_0(t)\| = 0, \quad \forall i \in \mathcal{V}.
\tag{7}
$$

## III. CONTROL DESIGN AND STABILITY ANALYSIS

In this section, we device a robust consensus protocol for a networked leader-following MAS subjected to distributed DoS attacks. The proposed protocol, leveraging dual-terminal DETM, ensures secure consensus of the system while optimizing control energy and conserving communication resources. Additionally, we present a thorough analysis of the protocol's stability and effectiveness.

### A. Design of Secure Consensus Protocol

The traditional control protocol (3) has been demonstrated to ensure asymptotically stable leader-following consensus in the absence of DoS attacks [15]. However, it may fail under cyber attacks. Moreover, this protocol necessitates continuous communication among agents and uninterrupted control updates from the control unit to the actuator, which is energy-intensive and challenging to implement in networked MASs.

To address this issue, we modify the conventional protocol by incorporating two DETMs into each agent's communicator and control-to-actuator channel, thereby eliminating the need

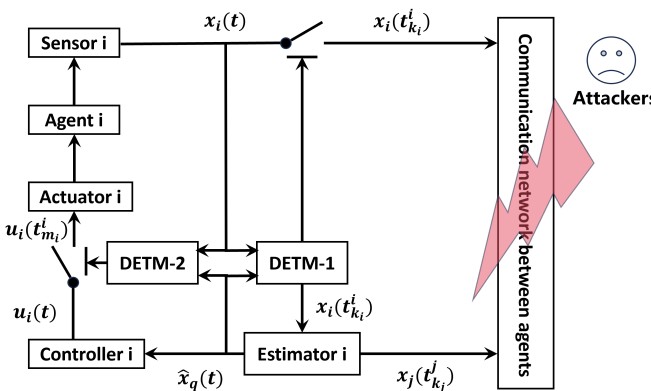

Fig. 2. Framework of the designed consensus protocol.

for continuous communication and control updates. Additionally, a secure hybrid update law is formulated to handle valid DoS attacks. The architecture of the proposed protocol is illustrated in Fig. 2.

Let $\{t_{k_i}^i\}_{k_i \in \mathbb{N}}$ be the set of time instants that agent $i$ transmits its state information to its neighbors, and denote the time instants that agent $i$ updates its control input as $\{t_{m_i}^i\}_{m_i \in \mathbb{N}}$, the control input of agent $i$ is given by the following distributed form:

$$
\begin{aligned}
u_i(t) &= K\hat{\vartheta}_i(t_{m_i}^i), \\
K &= -\upsilon R^{-1}B^T P, \\
\hat{\vartheta}_i(t) &= \sum_{j=1}^{N} a_{ij}(\hat{x}_i(t) - \hat{x}_j(t)) + d_i(\hat{x}_i(t) - \hat{x}_0(t)),
\end{aligned}
\tag{8}
$$

where $\upsilon$ is a positive constant to be determined later, $P > 0$ is the solution of the ARE $PA + A^T P - PBR^{-1}B^T P + Q = 0$ for two chosen matrices $Q > 0$ and $R > 0$, $\hat{x}_i(t)$, $\hat{x}_j(t)$ and $\hat{x}_0(t)$ are the estimates of $x_i(t)$, $x_j(t)$ and $x_0(t)$ by agent $i$, respectively, which can be generated by the following local estimator:

$$
\begin{cases}
\hat{x}_j(t_{k_j}^j) = x_j(t_{k_j}^j), & j \in \mathcal{N}_i(\tilde{\mathcal{G}}) \cup \{i\}, \\
\dot{\hat{x}}_j(t) = A\hat{x}_j(t), & t \in [t_{k_j}^j, t_{k_j+1}^j).
\end{cases}
\tag{9}
$$

Before preventing the dual-terminal DETM, we define the following two measurement errors:

$$
e_i(t) = x_i(t) - \hat{x}_i(t), \quad t \in [t_{k_i}^i, t_{k_i+1}^i), \tag{10}
$$

$$
e_{\vartheta_i}(t) = \hat{\vartheta}_i(t) - \hat{\vartheta}_i(t_{m_i}^i), \quad t \in [t_{m_i}^i, t_{m_i+1}^i), \tag{11}
$$

The triggering conditions of the designed dual-terminal DETM implemented on agent $i$ are presented as follows:

$$
DETM-1: t_{k_i+1}^i = \inf\{t > t_{k_i}^i | p_i(t) \geq 0\}, \tag{12}
$$

$$
p_i(t) = \|e_i(t)\|^2 - \alpha_{x_i}\|\hat{\vartheta}_i(t)\|^2 - \kappa_{x_i}\beta_{x_i}(t), \tag{13}
$$

$$
\dot{\beta}_{x_i}(t) = -\rho_{x_i}\beta_{x_i}(t) + \psi_{x_i}(-\|e_i(t)\|^2 + \alpha_{x_i}\|\hat{\vartheta}_i(t)\|^2), \tag{14}
$$

$$
DETM-2: t_{m_i+1}^i = \inf\{t > t_{m_i}^i | q_i(t) \geq 0\}, \tag{15}
$$

$$
q_i(t) = \|e_{\vartheta_i}(t)\|^2 - \alpha_{\vartheta_i}\|\hat{\vartheta}_i(t)\|^2 - \kappa_{\vartheta_i}\beta_{\vartheta_i}(t), \tag{16}
$$

$$
\dot{\beta}_{\vartheta_i}(t) = -\rho_{\vartheta_i}\beta_{\vartheta_i}(t) + \psi_{\vartheta_i}(-\|e_{\vartheta_i}(t)\|^2 + \alpha_{\vartheta_i}\|\hat{\vartheta}_i(t)\|^2), \tag{17}
$$

In addition, the initial value should be set as $\beta_{\vartheta_i}(0) > 0$, $\beta_{x_i}(0) > 0$.

As discussed in Section II-C, during the valid DoS attack interval $t \in \mathcal{D}(t_1, t_2)$, the protocol could be invalid due to the destroyed connectivity of the communication topology. To guarantee secure consensus under distributed DoS attacks occuring in the communication network, we design the following hybrid update law:

$$
t_{k_i+1}^i =
\begin{cases}
t_{k_i}^i + \tau_i, & \text{if } t_{k_i} \in \mathcal{D}(t_1, t_2), \\
DETM-1, & \text{if } t_{k_i} \in \mathcal{S}(t_1, t_2),
\end{cases}
\tag{18}
$$

$$
t_{m_i+1}^i =
\begin{cases}
t_{m_i}^i + \tau_i, & \text{if } t_{m_i} \in \mathcal{D}(t_1, t_2), \\
DETM-2, & \text{if } t_{m_i} \in \mathcal{S}(t_1, t_2),
\end{cases}
\tag{19}
$$

where $\tau_i > 0$ is a constant time interval to be chosen, indicating the agents perform periodic communication and control updates attempts during valid attack interval until the communication network is back up. In addition, for $t \in \mathcal{D}(t_1, t_2)$, this hybrid update law sets $\dot{\beta}_{x_i}(t)$, $\dot{\beta}_{\vartheta_i}(t)$ and $u_i(t)$ to zero to save control energy and computing resources.

Thus, the secure leader-following consensus protocol based on dual-terminal DETM is given by (8) to (19).

### B. Stability Analysis

In this subsection, we theoretically prove that the designed protocol can guarantee the asymptotic stability of the secure leader-following consensus.

Define the tracking error as $\tilde{x}_i(t) = x_i(t) - x_0(t)$. By combining (1)-(2) and (8)-(11) and letting $\tilde{x}(t) = \text{col}\{\tilde{x}_1(t), \dots, \tilde{x}_N(t)\}$, $e(t) = \text{col}\{e_1(t), \dots, e_N(t)\}$ and $e_\vartheta(t) = \text{col}\{e_{\vartheta_1}(t), \dots, e_{\vartheta_N}(t)\}$, the dynamics of the error system can be written in the following compact form:

$$
\begin{aligned}
\dot{\tilde{x}}(t) =& (I_N \otimes A + \mathcal{H} \otimes BK)\tilde{x}(t) - (\mathcal{H} \otimes BK)e(t) \\
& - (I_N \otimes BK)e_\vartheta(t).
\end{aligned}
\tag{20}
$$

To proceed with the stability analysis, let $b_{\tilde{x}} = \lambda_{min}(\Theta \otimes Q) = b_{\tilde{x}_1} + b_{\tilde{x}_2}$, $b_{\tilde{x}_1} > 0$, $b_{\tilde{x}_2} > 0$, $\Psi = PBR^{-1}B^T P$, $\mathcal{H}_I = \mathcal{H}^{-1} \otimes I_n$, $\Upsilon = \upsilon \mathcal{H}_I^T(\Theta \mathcal{H} \otimes \Psi) - b_{\tilde{x}_2}\mathcal{H}_I^T$, $\rho_{\tilde{x}} = b_{\tilde{x}_1}\lambda_{max}(\Theta \otimes P)^{-1}$, $b_\vartheta = \frac{3}{4}b_{\tilde{x}_2}\lambda_{min}(\mathcal{H}_I^T \mathcal{H}_I) - \frac{1}{a}\alpha_{max}(\Upsilon)$, $b_e = -b_{\tilde{x}_2} + \upsilon[\lambda_{max}(\Omega \otimes \Psi) + \lambda_{max}(\Theta \otimes \Psi)] + a\alpha_{max}(\Upsilon)$, $b_{e_\vartheta} = \frac{4c^2}{b_{\tilde{x}_2}}\lambda_{max}(\Theta \otimes \Psi)^2 + \upsilon\lambda_{max}(\Theta \otimes \Psi)$, where $a \in \mathbb{R}_{>0}$ such that $b_\vartheta > 0$ and $b_e > 0$, $\rho_1 = \min\{\rho_{\tilde{x}}, \rho_{x_i}, \rho_{\vartheta_i}\}$, $\rho_2 \in \mathbb{R}_{>0}$ satisfies $A^T Q + QA < \rho_2 Q$. Then, the following theorem summarizes the sufficient conditions for resilient consensus:

*Theorem 1:* Consider a networked MAS discussed in Section II-B subject to distributed DoS attack modeled in Section II-C. Assume the communication topology and DoS attacks satisfy Assumptions 1-3 hold, the protocol (8)-(19) is designed to guarantee secure leader-following consensus. If the following conditions are satisfied, the MAS can achieve secure consensus with asymptotic stability:

1) Condition on DETMs' parameters:

$$\rho_{x_i} > 0, \quad \rho_{\vartheta_i} > 0,$$
$$\psi_{x_i} > b_e, \quad \psi_{\vartheta_i} > b_{e_\vartheta}, \qquad (21)$$
$$\psi_{x_i}\alpha_{x_i} < b_{\vartheta_1}, \quad \psi_{\vartheta_i}\alpha_{\vartheta_i} < b_{\vartheta_2},$$

where $b_\vartheta = b_{\vartheta_1} + b_{\vartheta_2}$, $b_{\vartheta_1}$, $b_{\vartheta_2} \in \mathbb{R}_{>0}$.

2) Condition on control gain:

$$\upsilon \geq \frac{\lambda_{max}(\Theta)}{\lambda_{min}(\Omega)}. \qquad (22)$$

3) Constraints on attack duration and frequency:

$$T > \frac{\rho_1 + \rho_2}{\rho_1 - \rho^*}, \qquad (23)$$

and

$$\Gamma(0,t) < \frac{\rho_1}{2\ln\varrho + (\rho_1 + \rho_2)\tau_i}, \qquad (24)$$

where $\rho^* = 2\ln\varrho\Gamma(0,t) + (\rho_1 + \rho_2)\Gamma(0,t)\tau_i$, and $\varrho \in \mathbb{R}_{\geq 1}$ is a positive constant such that $\frac{1}{\varrho}Q \leq P \leq \varrho Q$.

*Proof:*

*Step 1. Classification of Intervals:*
The proposed secure hybrid update strategy (18)-(19) indicates that a DoS-induced time delay will occur when the valid attacks is over. Take the worst case into consideration, where the unhealthy interval $\mathcal{D}(t_1, t_2)$ is extended with $\tau_i$. Thus the time interval can be classified as:

$$\begin{cases} \mathcal{D}^*(t_1, t_2) = [t_1, t_2) \bigcap \bigcup_{n\in\mathbb{N}}[h_n, h_n + \tau_n + \tau_i), \\ \mathcal{S}^*(t_1, t_2) = [t_1, t_2)\backslash\mathcal{D}(t_1, t_2), \end{cases} \qquad (25)$$

*Step 2. Stability Analysis:*
1) When $t \in \mathcal{S}^*(t_1, t_2)$, $p_i(t) \leq 0$ and $q_i(t) \leq 0$ hold. Choose the Lyapunov function candidate as:

$$M_1(t) = V_1(t) + \sum_{i=1}^{N}(\beta_{x_i}(t) + \beta_{\vartheta_i}(t)), \qquad (26)$$

where

$$V_1(t) = \tilde{x}^T(t)(\Theta \otimes P)\tilde{x}(t). \qquad (27)$$

Taking the time-derivative of $M_i(t)$ and using (8), (22), and the ARE $PA + A^T P - PBR^{-1}B^T P + Q = 0$ yields

$$\dot{V}_1(t) \leq -b_{\tilde{x}}\tilde{x}^T(t)\tilde{x}(t) + 2c\tilde{x}^T(t)(\Theta\mathcal{H} \otimes \Psi)e(t) + 2c\tilde{x}^T(t)(\Theta \otimes \Psi)e_\vartheta(t), \qquad (28)$$

where $\Psi = PBR^{-1}B^T P$ and $b_{\tilde{x}} = \lambda_{min}(\Theta \otimes Q)$.

Refer to the definitions of $e_i(t)$ and $\hat{\vartheta}_i(t)$ and let $\hat{\vartheta}(t) = \text{col}\{\hat{\vartheta}_1(t), \ldots, \hat{\vartheta}_N(t)\}$, the following relationship can be deduced:

$$\tilde{x}(t) = \mathcal{H}_I\hat{\vartheta}(t) + e(t), \qquad (29)$$

where $\mathcal{H}_I = \mathcal{H}^{-1} \otimes I_n$.

Let $b_{\tilde{x}} = b_{\tilde{x}_1} + b_{\tilde{x}_2}$, where $b_{\tilde{x}_1}$ and $b_{\tilde{x}_2}$ be two positive constants. Then plugging (29) into (28) results in:

$$\begin{aligned} \dot{V}_1(t) \leq &-b_{\tilde{x}_1}\tilde{x}^T(t)\tilde{x}(t) - b_{\tilde{x}_2}\hat{\vartheta}^T(t)\mathcal{H}_I^T\mathcal{H}_I\hat{\vartheta}(t) \\ &-b_{\tilde{x}_2}e^T(t)e(t) + 2\hat{\vartheta}^T(t)\Upsilon e(t) \\ &+ ce^T(t)(\Omega \otimes \Psi)e(t) + 2c\hat{\vartheta}^T(t)\mathcal{H}_I^T(\Theta \otimes \Psi)e_\vartheta(t) \\ &+ 2ce^T(t)(\Theta \otimes \Psi)e_\vartheta(t), \end{aligned} \qquad (30)$$

where $\Upsilon = \upsilon\mathcal{H}_I^T(\Theta\mathcal{H} \otimes \Psi) - b_{\tilde{x}_2}\mathcal{H}_I$.

By applying Young's Inequality, the inequality (30) can be further derived as:

$$\begin{aligned} \dot{V}_1(t) \leq &-\rho_{\tilde{x}}\tilde{x}^T(t)(\Theta \otimes P)\tilde{x}(t) - b_\vartheta\|\hat{\vartheta}(t)\|^2 \\ &+ b_e\|e(t)\|^2 + b_{e_\vartheta}\|e_\vartheta(t)\|^2, \end{aligned} \qquad (31)$$

where $\rho_{\tilde{x}} = b_{\tilde{x}_1}\lambda_{max}(\Theta \otimes P)^{-1}$, $b_\vartheta = \frac{3}{4}b_{\tilde{x}_2}\lambda_{min}(\mathcal{H}_I^T\mathcal{H}_I) - \frac{1}{a}\alpha_{max}(\Upsilon)$, $b_e = -b_{\tilde{x}_2} + \upsilon[\lambda_{max}(\Omega \otimes \Psi) + \lambda_{max}(\Theta \otimes \Psi)] + a\alpha_{max}(\Upsilon)$, $b_{e_\vartheta} = \frac{4c^2}{b_{\tilde{x}_2}}\lambda_{max}(\Theta \otimes \Psi)^2 + \upsilon\lambda_{max}(\Theta \otimes \Psi)$, and $a \in \mathbb{R}_{>0}$ should be carefully chosen to maintain $b_\vartheta > 0$ and $b_e > 0$.

At last, if the conditions (21) are satisfied, the following inequality can be derived to hold over $t \in \mathcal{S}^*(t_1, t_2)$:

$$\begin{aligned} \dot{M}_1(t) \leq &-\rho_{\tilde{x}}\tilde{x}^T(t)(\Theta \otimes P)\tilde{x}(t) \\ &- \sum_{i=1}^{N}(\rho_{x_i}\beta_{x_i}(t) + \rho_{\vartheta_i}\beta_{\vartheta_i}(t)) \\ \leq &-\rho_1 M_1(t), \end{aligned} \qquad (32)$$

where $\rho_1 = \min\{\rho_{\tilde{x}}, \rho_{x_i}, \rho_{\vartheta_i}\}$.

2) When $t \in \mathcal{D}^*(t_1, t_2)$, $p_i(t) \leq 0$ and $q_i(t) \leq 0$ may not hold.

Choose the Lyapunov function candidate as

$$M_2(t) = V_2(t) + \sum_{i=1}^{N}(\beta_{x_i}(t) + \beta_{\vartheta_i}(t)), \qquad (33)$$

where

$$V_2(t) = \tilde{x}^T(t)(\Theta \otimes Q)\tilde{x}(t). \qquad (34)$$

Based on the designed secure hybrid update strategy, $u_i(t)$, $\dot{\beta}_{x_i}(t)$ and $\dot{\beta}_{\vartheta_i}(t)$ are set to 0 over $t \in \mathcal{D}^*(t_1, t_2)$. Consequently, taking the time-derivative of $V_2(t)$ yields

$$\begin{aligned} \dot{M}_2(t) = &\tilde{x}^T(t)[\Theta \otimes (A^T Q + QA)]\tilde{x}(t) \\ &< \rho_2 M_2(t), \end{aligned} \qquad (35)$$

where $\rho_2$ is a positive constant which satisfies $A^T Q + QA < \rho_2 Q$.

*Step 3. Conditions about Attack Frequency and Duration:*
The inequalities (32) and (35) deduced in the last step implies:

$$\begin{aligned} &\text{for } t \in [h_n, h_n + \tau_n + \tau_i): \\ &\qquad M_2(t) \leq e^{\rho_2(t-h_n)}M_2(h_n), \\ &\text{for } t \in [h_n + \tau_n + \tau_i, h_{n+1}): \\ &\quad M_1(t) \leq e^{-\rho_1(t-h_n-\tau_n-\tau_i)}M_1(h_n + \tau_n + \tau_i). \end{aligned} \qquad (36)$$

Let $\varrho \in \mathbb{R}_{\geq 1}$ be a gain scheduler that satisfies $\frac{1}{\varrho}Q \leq P \leq \varrho Q$ holds. Thus, we have $\frac{1}{\varrho}M_2(t) \leq M_1(t) \leq \varrho M_2(t)$ for $\forall t \in [0, \infty)$. Based on these discussions, without loss of generality, we have the following discussions over $t \in \mathcal{S}^*(0, \infty)$, :

$$\begin{aligned} M(t) \leq &e^{-\rho_1(t-h_n-\tau_n-\tau_i)}M_1(h_n + \tau_n + \tau_i) \\ \leq &\varrho e^{-\rho_1(t-h_n-\tau_n-\tau_i)}e^{\rho_2(\tau_n+\tau_i)}M_2(h_n) \\ \leq &\ldots \\ \leq &\varrho^{2\Lambda(0,t)}e^{-\rho_1(t-|\mathcal{D}^*(0,t)|)}e^{\rho_2|\mathcal{D}^*(0,t)|}M(0) \end{aligned} \qquad (37)$$

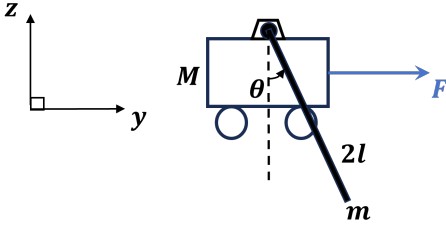

Fig. 3. Pendulum system for simulation

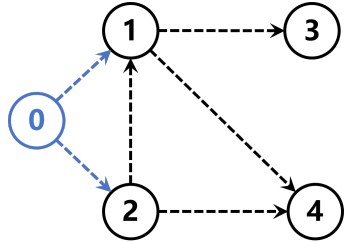

Fig. 4. Communication topology of the simulated MAS

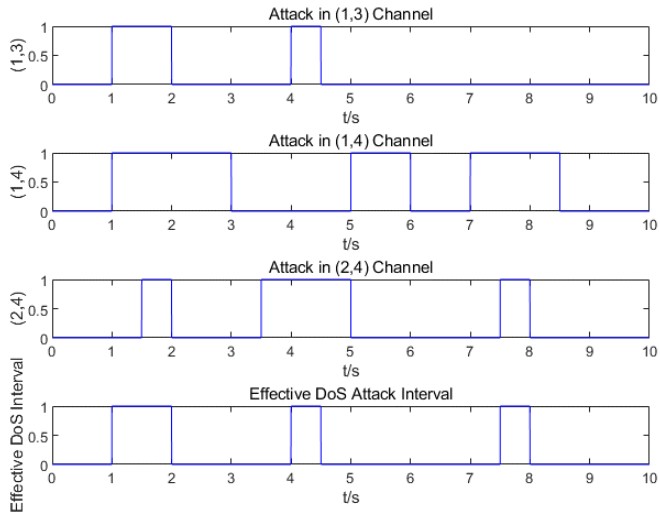

Fig. 5. Plot of distributed DoS attacks.

Consider the worst case of DoS-induced delay, in accordance with Assumption 2 and 3, the following inequality can be obtained:

$$|\mathcal{D}^*(0,t)| \leq \left(\delta_D + \frac{t}{T}\right) + (1 + \delta_F + \Gamma(0,t)t)\,\tau_i. \quad (38)$$

Plugging (38) into (37) results in

$$M(t) \leq e^{\Phi} e^{\Xi t} M(0), \quad (39)$$

where $\Phi = 2\delta_F \ln \varrho + (\rho_1 + \rho_2)[\delta_D + (1 + \delta_F)\tau_i]$, $\Xi = -\rho_1 + \frac{(\rho_1 + \rho_2)}{T} + \rho^*$, $\rho^* = 2\ln \varrho \Gamma(0,t) + (\rho_1 + \rho_2)\Gamma(0,t)\tau_i$. We have $\Xi < 0$ if conditions (23) and (24) are satisfied, which indicates $\lim_{t\to\infty} M(t) = 0$. Consequently, the secure leader-following consensus is guaranteed. ∎

## IV. NUMERICAL SIMULATION

In this section, we carry out a numerical simulation to verify the effectiveness of the proposed protocol.

Given a pendulum system shown in Fig. 3. Take the control input and state vector as $u_i(t) = F$ and $x = [y, \dot{y}, \theta, \dot{\theta}]^T$, respectively. In the simulation, we choose the linearized model around $u_i = 0$ and $x_i = 0_4$ as the agent dynamics. It is taken $m = 0.2kg$, $M = 0.5kg$, $l = 0.3m$ and $g = 9.81m/s^2$. It is easy to verified that the matrix pair $(A, B)$ of the agent model is stabilizable.

Consider a networked MAS with a communication topology as shown in Fig. 4, it can be verified that the topology includes a directed spanning tree with the leader agent 0 as the root node. For the simulation, the selected MAS and the designed protocol (8)-(19) are implemented in Simulink, with protocol parameters determined as discussed in Section III-B. The DoS attacks affecting various channels and the valid attack intervals are shown in Fig. 5. It can be observed that the graph's connectivity is disrupted over $t \in [1,2) \cup [4,4.5) \cup [7.5,8)$, so

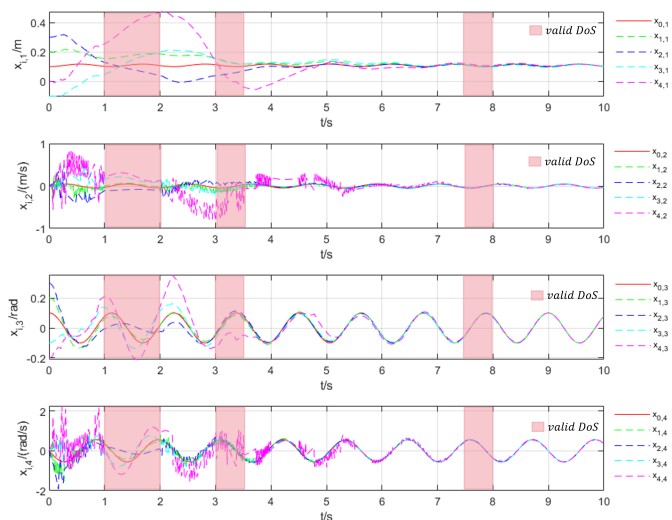

Fig. 6. Trajectory of states $x_i(t)$.

the conditions on attack duration and frequency (Assumptions 2-3) are satisfied. Fig. 6 plots the trajectories of all agents' states. The triggering times for control updates $t_{m_i}^i$ and communication $t_{k_i}^i$ for each agent are recorded in Fig. 7.

As shown in Fig. 6, even though the communication channels are asynchronously attacked, the followers can accurately track the leader's state. Thus, the secure leader-following consensus is achieved with asymptotic stability. As seen from Fig. 7, continuous control updates and communication are avoided throughout the entire consensus process, thereby conserving communication resources and control energy. Additionally, the record of triggering times indicates no agent exhibits Zeno behaviour.

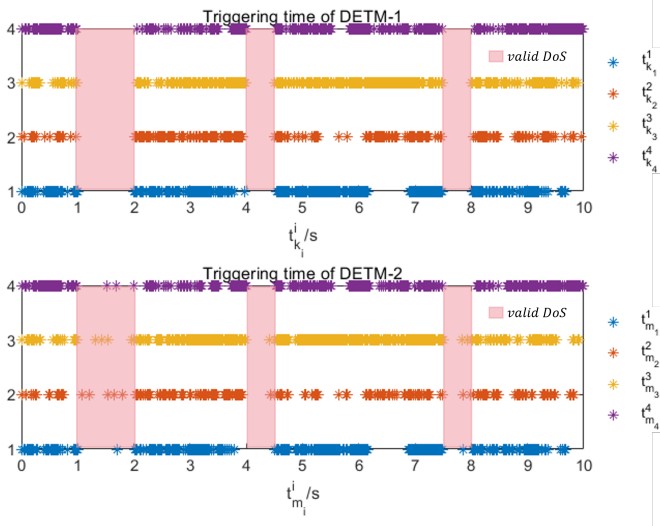

Fig. 7. Record of event times $t_{k_i}^i$ and $t_{m_i}^i$.

## V. CONCLUSION

In this paper, we explored the secure consensus problem for networked MASs under distributed DoS attacks using dual-terminal DETMs. We developed a distributed consensus protocol incorporating dual-terminal DETMs to ensure secure leader-following consensus. By implementing two DETMs on each agent, we effectively reduced control update and communication frequencies. Additionally, the secure hybrid update law, designed around the concept of valid DoS attack intervals, allows the system to recover from such attacks. Our theoretical analysis confirmed that secure leader-following consensus with asymptotic stability can be achieved if the attacks comply with specific duration and frequency constraints. Finally, simulation results demonstrated the effectiveness and validity of the proposed control strategies.

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
