# OpenReview forum: "Secure Dynamic Event-based Consensus for Networked Multi-agent Systems subject to Distributed DoS Attacks"
_IEEE.org/ICIST/2024/Conference — IEEE ICIST 2024 Conference Submission_

### Official Review · Reviewer_qAdR · 2024-08-24
**Review Comments for Manuscript No. 9**

**Rating:** 8
**Confidence:** 4

**Review:**

1. The distributed DoS attack's effect on the system is illustrated in Figure 2. Can the authors explain how distributed communication is reflected in this figure? Additionally, is there a contradiction between the communication mechanism of DETM-1 and the presence of DoS attacks depicted in the figure?

2. Regarding the structure of the agents, the main differences between followers and leaders are in the initial state values and the control laws. I have the following concerns:

   2.1. Apart from DoS attacks, followers do not seem to be influenced by disturbances or other signals. When matrix A is Hurwitz, the followers do not require control. If A is unstable, as long as the leader's state is captured at any moment, its future dynamics are predictable without further broadcasting. Given that the controller gain K in equation (3) is the same for all agents, if the state of the agent connected to node i is detected at some point, can subsequent communication be avoided?

   2.2. In the simulation section, the authors mention \(x_i = 0_4\). Does this mean that all initial states are the same? If so, does this contradict the simulation results shown in Figure 6? Additionally, what does \(u_i = F\) refer to?

3. What are the advantages of the proposed method compared to existing pinning control design methods under DoS attacks or varying communication topologies in complex dynamical networks?

4. How does the designed event-triggered mechanism avoid the Zeno phenomenon? What is the scalability of the proposed control law?

5. There are some formatting errors in the references. The authors should make the necessary corrections.

---

### Official Review · Reviewer_aiJS · 2024-08-29
**This paper can be accepted.**

**Rating:** 10
**Confidence:** 5

**Review:**

This paper addresses the secure leader-following consensus problem in networked multi-agent systems (MASs) subjected to distributed DoS attacks. A secure hybrid update law to maintain secure consensus under distributed DoS attacks is given. Theoretical analysis confirms that the proposed protocol guarantees asymptotically stable leaderfollowing consensus, given that the duration and frequency of valid attacks are constrained, and no agent exhibits Zeno behavior.

This paper is well organized and contains unique contributions. The derivations seem correct. The simulation results show the effectiveness and advantages of the proposed method.

---

### Official Review · Reviewer_yohS · 2024-09-03
**This paper is well organized and can be accepted.**

**Rating:** 9
**Confidence:** 3

**Review:**

The authors in this paper investigate the secure dynamic event-based consensus control problem for networked multi-agent systems subject to distributed DoS attacks. The reviewer has the following comments for this paper.
1. Please add a remark to show the guidelines for parameter selecting.
2. Some future investigation topics should be mentioned in the Conclusion part.

---

### Decision · Program_Chairs · 2024-09-06

Accept (Oral)